# Exploring the Potential of High-Molar-Activity Samarium-153 for Targeted Radionuclide Therapy with [^153^Sm]Sm-DOTA-TATE

**DOI:** 10.3390/pharmaceutics14122566

**Published:** 2022-11-23

**Authors:** Koen Vermeulen, Michiel Van de Voorde, Charlotte Segers, Amelie Coolkens, Sunay Rodriguez Pérez, Noami Daems, Charlotte Duchemin, Melissa Crabbé, Tomas Opsomer, Clarita Saldarriaga Vargas, Reinhard Heinke, Laura Lambert, Cyril Bernerd, Andrew R. Burgoyne, Thomas Elias Cocolios, Thierry Stora, Maarten Ooms

**Affiliations:** 1NURA Research Group, Belgian Nuclear Research Center (SCK CEN), 2400 Mol, Belgium; 2Institute for Nuclear and Radiation Physics, KU Leuven, 3000 Leuven, Belgium; 3MEDICIS, Conseil Européen pour la Recherche Nucléaire (CERN), 1211 Geneva, Switzerland; 4Research in Dosimetric Applications, Belgian Nuclear Research Center (SCK CEN), 2400 Mol, Belgium

**Keywords:** targeted radionuclide therapy, samarium-153, DOTA-TATE, SSTR_2_

## Abstract

Samarium-153 is a promising theranostic radionuclide, but low molar activities (A_m_) resulting from its current production route render it unsuitable for targeted radionuclide therapy (TRNT). Recent efforts combining neutron activation of ^152^Sm in the SCK CEN BR2 reactor with mass separation at CERN/MEDICIS yielded high-A_m_ ^153^Sm. In this proof-of-concept study, we further evaluated the potential of high-A_m_ ^153^Sm for TRNT by radiolabeling to DOTA-TATE, a well-established carrier molecule binding the somatostatin receptor 2 (SSTR_2_) that is highly expressed in gastroenteropancreatic neuroendocrine tumors. DOTA-TATE was labeled with ^153^Sm and remained stable up to 7 days in relevant media. The binding specificity and high internalization rate were validated on SSTR_2_-expressing CA20948 cells. In vitro biological evaluation showed that [^153^Sm]Sm-DOTA-TATE was able to reduce CA20948 cell viability and clonogenic potential in an activity-dependent manner. Biodistribution studies in healthy and CA20948 xenografted mice revealed that [^153^Sm]Sm-DOTA-TATE was rapidly cleared and profound tumor uptake and retention was observed whilst these were limited in normal tissues. This proof-of-concept study showed the potential of mass-separated ^153^Sm for TRNT and could open doors towards wider applications of mass separation in medical isotope production.

## 1. Introduction

Samarium-153 is a radioisotope that has been used in clinical practice for many years because of its favorable decay properties. ^153^Sm has a half-life of 1.93 days and decays into a stable daughter nuclide (^153^Eu). Upon its decay, several β-particles (E = 808 keV (18%), 705 keV (50%), 635 keV (32%)) and Auger–Meitner and conversion electrons are emitted which are responsible for the therapeutic potential of ^153^Sm. The average β-energy for ^153^Sm is 91 keV/particle higher than the current benchmark ^177^Lu with almost six times more Auger–Meitner and conversion electrons per nuclear transformation (Table 1) [1]. In parallel, ^153^Sm also co-emits γ-rays (103 keV, 28%) that allow single photon emission computed tomography (SPECT) imaging using clinically available collimators [2]. These properties put ^153^Sm forward as a promising theranostic radionuclide for targeted radionuclide therapy (TRNT).

Despite its high potential in TRNT, the use of ^153^SmCl_3_ (as Quadramet^®^) is currently restricted to bone pain palliation for patients suffering from bone metastases originating from various types of cancer [3,4]. The limited applicability of ^153^Sm mainly originates from its current production method. ^153^Sm is efficiently produced by neutron activation using the high thermal neutron flux of a nuclear research reactor, i.e., via the ^152^Sm (*n*,γ) ^153^Sm nuclear reaction [5]. The produced ^153^Sm cannot be chemically separated from its irradiation matrix, resulting in low molar activities (A_m_) that are unsuitable for TRNT.

Exploiting the potential of ^153^Sm for TRNT will only be possible if the A_m_ of ^153^Sm can be significantly increased. In an earlier collaborative study, we successfully combined neutron activation of ^152^Sm in the BR2 reactor with mass separation at CERN/MEDICIS [6]. Doing this, we were able to drastically increase the A_m_ of the ^153^Sm, allowing for high-A_m_ radiolabeling of chelator molecules. Initial radiolabeling studies were performed; however, applicability of the ^153^Sm in TRNT has not yet been shown.

Building further on our previous work [6], we aimed to prove the potential of ^153^Sm in TRNT in this study. High-A_m_ ^153^Sm was used to radiolabel DOTA-TATE, a well-established carrier molecule that specifically binds the somatostatin receptor 2 (SSTR_2_) that is highly expressed in gastroenteropancreatic neuroendocrine tumors (GEP-NETs). The development, evaluation and clinical translation of this compound pioneered the development of TRNT [7]; hence, it is often used as a platform to expand TRNT into other cancers/targets. Therefore, in this proof-of concept study, DOTA-TATE was used to evaluate the applicability of ^153^Sm in TRNT. In this manuscript, we report the radiolabeling and early biological evaluation of [^153^Sm]Sm-DOTA-TATE in SSTR_2_-expressing CA20948 cells and tumor models of pancreatic neuroendocrine cancers.

**Table 1 pharmaceutics-14-02566-t001:** Comparison of decay properties of ^177^Lu and ^153^Sm.

	^177^Lu	^153^Sm
Half-life (days)	6.7	1.9
β-yield (particle/nt)	1	1
β-energy	498 (79%) 385 (9%) 176 (12%)	808 (18%) 705 (50%) 635 (32%)
β-energy (keV/nt)	133	224
β-av. energy (keV/particle)	133	224
IC electrons yield (particle/nt)	0.15	0.81
IC electrons energy (keV/nt)	14	40
IC electrons av. energy (keV/particle)	87	50
AM electrons yield (particle/nt)	1.12	6.58
AM electrons energy (keV/nt)	1	6
AM electrons av. energy (keV/particle)	1	0.9
AM and IC electrons yield (particle/nt)	1.27	7.38

Table adapted with permission from Uusijärvi et al. [1] and IRCP Publication 107—Nuclear Decay Data for Dosimetry Calculations [8]. Nt = nuclear transformation, IC = internal conversion, AM = Auger–Meitner.

## 2. Materials and Methods

### 2.1. Chemicals

All chemicals were purchased from Sigma-Aldrich (Bornem, Belgium) or VWR (Leuven, Belgium), and were used without further purification. DOTA-TATE (DOTA-[Tyr3]octreotate) was purchased from ABX (Radeberg, Germany).

### 2.2. Quantification of Radioactivity in Biological Samples

Quantification of activity was performed with an automated gamma counter equipped with a 3-in NaI(Tl) detector coupled to a multichannel analyzer (2480 Wizard^2^_,_ Perkin Elmer, Zaventem, Belgium). The results were corrected for background signal, physical decay during counting and detector dead time.

### 2.3. Cell Culture

Rat pancreatic cancer cell line CA20948 was used given the somatostatin receptor 2 (SSTR_2_) overexpression as it is found in many human neuroendocrine tumors [9]. Cells were grown in DMEM High Glucose and Pyruvate medium (Gibco 41966029) supplemented with 10% fetal bovine serum (Gibco 10270106) and 100 U/mL penicillin-streptomycin (Sigma-Aldrich P4333) and were maintained in a humidified incubator at 37 °C with 5% CO_2_ supply. All in vitro results were obtained from a minimum of three replicates per condition.

### 2.4. Mice

All animal experiments were performed in compliance with the Ethical Committee Animal Studies of Medanex Clinic (EC MxCl 2020-163 and EC MxCl 2021-173), the Belgian laboratory animal legislation and the European Communities Council Directive of 22 September 2010 (2010/63/EU). Male BALB/c and NMRI nude mice were purchased from Janvier (Bio Services, Uden, The Netherlands) and housed in ventilated cages under standard laboratory conditions (12 h light/dark cycle) at the animal facility of SCK CEN. All animals had access to food and water ad libitum.

### 2.5. Generation of Tumor Bearing Xenograft Mice

Male NMRI nude mice (between 8 and 15 weeks old) were anesthetized with 2.5% isoflurane in O_2_ at a flow rate of 0.2 L min^−1^ and subcutaneously injected on the right shoulder with 1 × 10^6^ CA20948 cells resuspended in PBS with 30% matrigel^®^. Tumor dimensions were measured two or three times per week and volume was calculated as length × width × height. Mice were sacrificed if the tumor volume exceeded 600 mm^3^ or if more than 10% of the original body weight was lost.

### 2.6. Samarium-153 Production

Samarium-153 was produced as described in our previous work [6]. In short, ^153^Sm was produced by neutron activation of highly enriched ^152^Sm in its nitrate form (enrichment grade 98.7%). About 350 µg ^152^Sm was sealed in an ampoule and irradiated in the BR2 reactor for 2 or 3 days in thermal neutron fluxes of 2.0 to 2.5 × 10^14^ neutrons/cm²/s. Up to three ampoules were irradiated in each production. After cooling, the target material was dissolved and shipped to CERN/MEDICIS for mass separation to yield high-A_m_ ^153^Sm. After mass separation, the ^153^Sm was radiochemically purified to remove any ^153^Eu and other impurities arising from the mass separation process (e.g., Zn from the implantation layer), resulting in high-purity [^153^Sm]SmCl_3_ (in 0.05 M HCl) ready for radiolabeling.

### 2.7. Radiolabeling

Radiolabeling was performed by adding up to 200 MBq of the ^153^Sm radiolabeling solution to 20 nmol of the DOTA-TATE precursor dissolved in 0.15 M NaOAc buffer (pH 4.7) in a total reaction volume of 1 mL. The reaction vial was placed in a Thermomixer C (Eppendorf) to stir (500 rpm) and the mixture was heated to 95 °C for 15 min. After incubation, the radiolabeling yield was evaluated using thin-layer chromatography. An amount of 1–2 µL of the reaction mixture was spotted on a glass microfiber chromatography paper strip impregnated with silica gel (iTLC-SG, Agilent Technologies, Diegem, Belgium) that was eluted with a 0.5 M citrate solution (pH 5.5). After elution, the TLC papers were cut in half and the activity of the bottom and top part of the TLC paper was counted for 2 min each using an automated gamma counter.

### 2.8. Stability Test

The stability of [^153^Sm]Sm-DOTA-TATE was evaluated by incubating the radiopharmaceutical in several relevant media. About 5 MBq of [^153^Sm]Sm-DOTA-TATE (250 µL) was added to 4.75 mL of human serum or 4.75 mL of PBS and incubated at 37 °C up to 7 days. After 1 h, 3 days, 5 days and 7 days, a 5 µL sample was spotted on the TLC system described above to identify the intact fraction. To get an idea of the shelf life of [^153^Sm]Sm-DOTA-TATE, the stability of the radiopharmaceutical product was evaluated in radiolabeling buffer at room temperature using the same protocol.

### 2.9. Cell Binding Assay

To study membrane association and internalization of [^153^Sm]Sm-DOTA-TATE, 50,000 cells were seeded per well in a 24-well plate (26,316 cells/cm²). After 24 h, cells were incubated with 2.5 nM [^153^Sm]Sm-DOTA-TATE (2.7 kBq) for 1 h. Non-specific binding was assessed by co-incubation of [^153^Sm]Sm-DOTA-TATE with a 4-fold and 4000-fold molar excess of unlabeled DOTA-TATE in parallel. Hereafter, fractions were collected. First, the membrane-bound fraction was collected following 10 min incubation at room temperature in strip buffer (50 mM glycine, 100 mM NaCl, pH 2.8). Then, the internalized fraction was collected by 30 min of cell lysis (1 M NaOH) at room temperature. The amount of [^153^Sm]Sm-DOTA-TATE associated with the cell membrane and internalized was measured using an automated gamma counter. In addition, total cell number per well was obtained using a MOXI™ automated cell counter (VWR, Leuven, Belgium).

### 2.10. MTS Viability Assay

Cell viability after exposure to [^153^Sm]Sm-DOTA-TATE was tested using the CellTiter 96^®^ Aqueous One Solution Cell Proliferation Assay (Promega G3582). A total of 10,000 cells were seeded per well into a 96-well plate (31,250 cells/cm²). After 24 h, cells were treated with [^153^Sm]Sm-DOTA-TATE (0–10 MBq/mL) in fresh cell culture medium. After 4 h incubation at 37 °C in 5% CO_2_, cells were washed and new medium was added. Then, after 4 days, MTS reagent was prepared according to the manufacturer’s instructions and added to the cells. Absorbance was measured at 490 nm using the BioTek Synergy H1 reader (Agilent). Cell viability was obtained as the ratio of the absorbance of a sample well to the average absorbance of untreated (0 MBq/mL) wells, after subtraction of the background signal of wells containing MTS reagent without cells. Statistical significance (*p* < 0.05) was determined using linear models.

### 2.11. Clonogenic Survival Assay

Cell suspensions (0.1 × 10^6^ cells) were prepared using 0.25% trypsin/EDTA, resuspended in cell culture medium and treated for 4 h with radiolabeled [^153^Sm]Sm-DOTA-TATE (0–10 MBq/mL). An aliquot of 300 cells for each treatment condition was plated in 6-well plates with 2 mL of growth medium and incubated at 37 °C in 5% CO_2_. After 14 days, colonies were fixed and stained with 6% glutaraldehyde and 0.5% crystal violet. The number of colonies (blindly coded) was counted to determine the clonogenic survival fraction per treatment condition. Importantly, colonies were only considered if they contained at least 50 cells. Statistical significance (*p* < 0.05) was determined using linear models.

### 2.12. Biodistribution

The pharmacokinetic behavior of [^153^Sm]Sm-DOTA-TATE was evaluated by performing biodistribution studies in healthy BALB/c mice as well as in CA20948 tumor bearing xenograft mice [9]. Additionally, similar biodistribution studies were performed with free (unlabeled) [^153^Sm]SmCl_3_ to identify the organs with ^153^Sm accumulation in case of release from the complex. All animals were anesthetized with 2.5% isoflurane in O_2_ at a flow rate of 1 L min^−1^ and injected with ∼2–3 MBq of [^153^Sm]SmCl_3_ or [^153^Sm]Sm-DOTA-TATE via a tail or penile vein. The mice were sacrificed by an overdose pentobarbital (200 µL of 60 mg/mL) 1, 4, 24 and 72 h post-injection (p.i.) (*n* = 3–4 per time point). Blood was collected via cardiac puncture. Organs of interest were collected in tared tubes and weighed. The activity in the different organs was measured in the gamma counter. For the calculation of the total activity in the blood, bone and muscle, the masses were assumed to be, respectively, 7, 12 and 40% of the total body mass [10,11]. Data were expressed as a percentage of injected activity (%IA) and standardized uptake value (SUV). %IA was calculated as (counts per min (cpm) in organ)/(IA converted to cpm) × 100. SUV was calculated as (activity in cpm in organ/weight of organ in g)/(IA/total body weight in g).

### 2.13. Dosimetry

The mean absorbed doses per amount of injected activity (mGy/mBq) to the kidneys and tumor were calculated based on the MIRD formalism. The time–activity curve (TAC) was created based on the fraction of injected activity per gram of dissected tissue (FIA/g) as a function of time. The FIA was decay-corrected to the time of sacrifice of the mouse. For both tissues, it was assumed that FIA/g = 0 at t = 0 h (injection time) and to increase linearly thereafter until t = 1 h (earliest time point). The time–activity curves for kidneys and tumor were fitted to bi-exponential curves (R^2^ > 0.99 and R^2^ = 0.97, respectively), which were then integrated from zero to infinity. The time-integrated activity coefficient (ã) was obtained by multiplying the mean tissue mass (i.e., m = 0.42 g for tumor and m = 0.26 g for kidney) by the respective time-integrated activity.

The mean absorbed dose for tumor and kidney was obtained by multiplying ã and the *S* values (0.0962 and 0.328 mGy/MBq.s for tumor and kidney, respectively). The *S* values were calculated for ^153^Sm radiation decay data using the Monte Carlo code MCNP6.2 (Los Alamos National Laboratory, Los Alamos, NM, USA). Spectrum data from ^153^Sm were modelled according to the decay data reported in ICRP Publication 107 [8]. A water sphere of 0.42 g (for the tumor) and a realistic model of a mouse kidney [12] were used to model the tissue geometry in the Monte Carlo simulations. Additionally, the *S* value for kidneys was scaled to the mean measured kidney tissue mass.

### 2.14. SPECT Imaging

To evaluate the imaging possibilities of [^153^Sm]Sm-DOTA-TATE, one CA20948 tumor xenograft mouse was subjected to a SPECT scan. The mouse was anaesthetized using isoflurane (2.5% in O_2_, flow rate 1 L/min) and injected with about 20 MBq of [^153^Sm]Sm-DOTA-TATE via the tail vein. At 4 h and 24 h after injection, the mouse was scanned for approximately 2 h, followed by a 3 min CT scan. The in vivo imaging was performed in a dedicated small animal SPECT-CT (U-SPECT6-CT, Milabs, Utrecht, The Netherlands). The scans were acquired in list-mode using a general-purpose mouse collimator with 0.6 mm diameter pinholes.

The images were reconstructed and co-registered using the Milabs provided manufacturer software. SPECT images were reconstructed using a 20% energy window centered on the 103 keV photopeak. Scatter correction was applied using the triple energy window method with two adjacent background-scatter windows of 4.2% width. SPECT image reconstruction was performed using the similarity-regulated ordered-subsets expectation maximization (SROSEM) algorithm [13], with four iterations, 128 subsets and a voxel size of 0.4 mm^3^.

SPECT images were registered to the CT images, for a final voxel size of 0.16 mm^3^, and corrected for photon attenuation based on the CT data. Data post-processing and analysis was performed using PMOD 4.101 software (PMOD, Zürich, Switzerland).

SPECT data in reconstructed counts per seconds (cps) units were calibrated to activity units (MBq/cc) using a calibration factor (CF). The CF was calculated from a SPECT scan of a syringe filled with a known activity concentration of [^153^Sm]SmCl_3_. The activity concentration was traceable to high-resolution gamma spectrometry measurements performed on-site.

## 3. Results

### 3.1. Radiolabeling and Stability

Radiolabeling of [^153^Sm]Sm-DOTA-TATE at 95 °C was highly efficient, resulting in high radiochemical conversion rates (>99%) with an A_m_ of up to 10 MBq/nmol. The shelf life of [^153^Sm]Sm-DOTA-TATE was tested by storing the radiopharmaceutical at room temperature in its radiolabeling buffer. Up to 7 days after incubation in radiolabeling buffer, no degradation of the [^153^Sm]Sm-DOTA-TATE complex was observed (radiochemical purity > 98%) (Figure 1). Additionally, incubating [^153^Sm]Sm-DOTA-TATE in biologically relevant media like PBS and human serum at 37 °C did not result in any dissociation of ^153^Sm from the DOTA-complex (radiochemical purity > 99%).

### 3.2. In Vitro Specificity and Cellular Internalization of [^153^Sm]Sm-DOTA-TATE

The specific binding of [^153^Sm]Sm-DOTA-TATE to SSTR_2_ receptors was evaluated in CA20948 cells. After 1 h of incubation, 0.3% of the added activity was associated with the CA20948 cells, of which 79 ± 2% was internalized (Figure 2). Co-incubation of, respectively, a 4- and a 4000-fold excess of unlabeled DOTA-TATE drastically decreased the binding, with about 40% and 92%, proving the binding specificity.

### 3.3. In Vitro Biological Evaluation of [^153^Sm]Sm-DOTA-TATE

Cell viability was assessed after exposure to increasing concentrations of [^153^Sm]Sm-DOTA-TATE. MTS assays revealed that [^153^Sm]Sm-DOTA-TATE treatment significantly reduced cell viability down to 48 ± 6% in an activity concentration-dependent manner when compared to untreated cells (Figure 3). Notably, treatment with [^153^Sm]Sm-DTPA also significantly lowered cell viability down to 45 ± 5% in an activity concentration-dependent manner when compared to untreated cells (Figure 3). However, no effect on cell viability (85 ± 10%) was observed with the lowest tested activity concentration (1.25 MBq/mL) of [^153^Sm]Sm-DTPA in comparison to untreated cells (100 ± 17%). When comparing both treatments, a higher impact on viability (*p* = 0.02) was observed for 5 MBq/mL [^153^Sm]Sm-DTPA in comparison to 5 MBq/mL [^153^Sm]Sm-DOTA-TATE.

In addition, the clonogenic potential of CA20948 cells was evaluated following [^153^Sm]Sm-DOTA-TATE treatment. In line with the viability assay, both treatments with [^153^Sm]Sm-DOTA-TATE and [^153^Sm]Sm-DTPA significantly decreased the surviving fraction of cells (down to 28 ± 16% for 10 MBq [^153^Sm]Sm-DOTA-TATE and 13 ± 5% for 10 MBq [^153^Sm]Sm-DTPA) in a concentration-dependent fashion compared to untreated cells (100 ± 23% for 0 MBq [^153^Sm]Sm-DOTA-TATE and 100 ± 11% for 0 MBq [^153^Sm]Sm-DTPA) (Figure 4). A significant difference in the surviving fraction (*p* = 0.01) was only observed for 2.5 MBq/mL [^153^Sm]Sm-DTPA in comparison to 2.5 MBq/mL [^153^Sm]Sm-DOTA-TATE.

### 3.4. Biodistribution Studies

The biodistribution of [^153^Sm]Sm-DOTA-TATE in healthy BALB/c mice showed rapid excretion of radioactivity from the body (Appendix A). After 1 h, already 82.0 ± 4.8% of the activity was cleared from the body. After 72 h, the excreted fraction increased to 97.8 ± 0.2 %IA. The initial uptake of [^153^Sm]Sm-DOTA-TATE in kidneys (SUV_1h_ = 2.1 ± 0.2; SUV_4h_ = 1.8 ± 0.1) and a limited liver and intestinal uptake suggest renal clearance (Figure 5 and Appendix A). Apart from some initial kidney uptake, only slight retention of radioactivity was observed in spleen.

Unlabeled [^153^Sm]SmCl_3_, on the other hand, showed high accumulation in liver and bone tissue (SUV_72h_ = 7.7 ± 1.8 and 3.1 ± 1.3 for liver and bone, respectively) (Appendix A). This contrasts with the limited bone uptake observed in [^153^Sm]Sm-DOTA-TATE-injected animals (SUV_72h_ = 0.1 ± 0.0, Figure 5 and Appendix A).

In order to evaluate the pharmacokinetic profile in tumor bearing mice, [^153^Sm]Sm-DOTA-TATE was injected in CA20948 xenograft mice. Again, rapid excretion was observed with about 80 %IA already excreted after 1 h p.i. (Appendix A). At the initial time points, mild uptake was observed in kidneys (SUV_1h_ = 1.8 ± 0.1) and pancreas (SUV_1h_ = 1.2 ± 0.2), which rapidly washed out from these organs (Figure 6 and Table 2). Substantial tumor uptake was observed 1 h p.i. (SUV_1h_ = 4.5), which only slowly decreased over the different time points (SUV_72h_ = 1.7). The limited kidney SUV uptake and good retention in the tumor resulted in high tumor-to-kidney SUV ratios (Table 2). The lack of uptake in bone tissue (SUV_72h_ = 0.0 ± 0.0) again confirms the in vivo stability of the [^153^Sm]Sm-DOTA-TATE complex.

### 3.5. Dosimetry

The absorbed dose estimated for a spherical tumor (m = 0.42 g) was 30.7 mGy/MBq·s. This results in 6 mGy/s dose to the tumor for a mouse injected with 2 MBq of [^153^Sm]Sm-DOTA-TATE. For the kidneys, the absorbed dose was 3.35 mGy/MBq·s, which results in 6.7 mGy/s dose to the kidneys for a mouse injected with 2 MBq of [^153^Sm]Sm-DOTA-TATE.

### 3.6. SPECT-CT Imaging

Small-animal SPECT-CT scans were acquired at 4 and 24 h p.i. of ~20 MBq of [^153^Sm]Sm-DOTA-TATE in CA20948 tumor-bearing mice resulting in the images shown in Figure 7. The images are displayed as maximum intensity projection (MIP) and are expressed in units of MBq/cc. In the figure, the tumor (Tu) can be observed in the right shoulder of the mouse. Other organs visible in the image are kidneys (Ki), liver (Li) and bladder (Bl).

[^153^Sm]Sm-DOTA-TATE showed a time-dependent tumor uptake of 1.01 MBq/cc and 0.56 MBq/cc at 4 and 24 h p.i., respectively. Rapid renal clearance is observed, with the bladder showing a high accumulation of activity at 4 h p.i. which has mostly cleared at 24 h p.i.. Other organs observed in the SPECT-CT images were kidneys and liver, the latter with limited retention of activity. Additionally, no bone uptake was observed. The findings on the in vivo imaging are in line with the biodistribution data.

## 4. Discussion

Based on its favorable decay characteristics, ^153^Sm can be a valuable radionuclide for targeted radionuclide therapy (TRNT). However, due to the current standard production route, only low-A_m_ ^153^Sm can be obtained, rendering it unsuitable for TRNT. This changed when the Belgian Nuclear Research Centre and the MEDICIS research branch of CERN joined forces to produce high-A_m_ ^153^Sm. In a previous study, we successfully used mass separation in combination with laser resonance ionization to extract ^153^Sm from its irradiation matrix resulting in non-carrier-added, high-A_m_ ^153^Sm [6]. Initial radiolabeling studies showed that ^153^Sm could be incorporated in *p*-SCN-Bn-DOTA with high yields. The applicability of the produced high-A_m_ ^153^Sm in TRNT was yet to be proven. Therefore, this proof-of-concept study aimed to investigate the potential of ^153^Sm for TRNT. DOTA-TATE was chosen as the preferred vector for the evaluation of the potential of ^153^Sm in TRNT. The FDA-approved [^177^Lu]Lu-DOTA-TATE (Luthatera^®^) has a proven effect on the progression-free survival of patients suffering from GEP-NETs [7]. With the development of the high-A_m_ production method, ^153^Sm could be added to the group of radionuclides used in TRNT. In a clinical study comparing ^153^Sm-EDTMP and ^177^Lu-EDTMP for bone pain palliation, it was observed that both radionuclides were equally effective and, more importantly, could be used interchangeably [14]. Compared to ^177^Lu, ^153^Sm could lead to improved therapeutic efficacy for TRNT of small and medium-to-large tumors. This is attributed to the emission of higher-energy β^−^ particles and a significant amount of Auger–Meitner and conversion electrons (Table 1) [1]. This high Auger–Meitner decay, combined with a shorter half-life and emission of gamma-rays, makes ^153^Sm a versatile radioisotope. Efforts are currently being made to scale up the production of high-A_m_ ^153^Sm.

We successfully radiolabeled [^153^Sm]Sm-DOTA-TATE at fair A_m_ that showed no leaching of ^153^Sm from the DOTA complex for at least 7 days. This was in line with earlier data showing that DOTA forms a very stable complex with several lanthanides, including ^153^Sm [15]. The stability observed in vitro was also confirmed in our in vivo results. Our data showed that ^153^SmCl_3_ highly accumulated in bone and liver. When injecting [^153^Sm]Sm-DOTA-TATE, these organs did not show any uptake in the SPECT images and biodistribution studies. Therefore, we can safely assume that the complex between ^153^Sm and DOTA-TATE is highly stable both in vitro and in vivo.

Our cell binding experiments showed specific binding to the SSTR_2_-expressing CA20948 cells. Both CA20948 and SSTR_2_-transfected U2OS cells are routinely used in preclinical studies concerning cancers overexpressing SSTR_2_ [16,17,18]. CA20948 cells are derived from rat pancreas and express SSTR_2_ endogenously [18]. These cells can be easily used to generate xenografts, whereas this is not the case for SSTR_2_-transfected U2OS cells. Clear and specific binding of [^153^Sm]Sm-DOTA-TATE to the CA20948 cells was observed, of which the largest part was internalized. These data are in line with Dalm et al. [19]. The absolute uptake in our study was lower compared to the reported literature in the same cell line, which could be attributed to the lower A_m_ used in our experiments. In parallel, the comparison of uptake between [^177^Lu]Lu-DOTA-TATE and lower A_m_ [^213^Bi]Bi-DOTA-TATE also yielded roughly a 100-fold-lower uptake of the low-A_m_ compound [20].

An activity-dependent effect of [^153^Sm]Sm-DOTA-TATE was observed on CA20948 cell viability and clonogenic potential. In parallel, we performed the same experiment with [^153^Sm]Sm-DTPA at the same radioactive concentrations, as a surrogate for radioactive effects originating from the medium. DTPA is commonly used since it easily complexes lanthanides and is not expected to show any affinity towards CA20948 cells. Although the uptake of [^153^Sm]Sm-DTPA in the CA20948 cells was not studied, the literature results testing the binding of [^177^Lu]Lu-DTPA showed a complete lack of affinity towards SSTR_2_ in CA20948 cells [20,21]. As such, we assumed that, upon changing the radionuclide, [^153^Sm]Sm-DTPA would also not be internalized by CA20948 cells. In the study by Chan et al. [20], the clonogenic potential of CA20948 cells after exposure to [^177^Lu]Lu-DOTA-TATE and [^177^Lu]Lu-DTPA, as well as [^213^Bi]Bi-DOTA-TATE and [^213^Bi]Bi-DTPA, were compared. In this study, despite the absence of an impact by [^177^Lu]Lu-DTPA when compared to [^177^Lu]Lu-DOTA-TATE, a similar cytotoxic effect was observed for [^213^Bi]Bi-DOTA-TATE and [^213^Bi]Bi-DTPA in CA20948 cells, indicating that indeed a cytotoxic dose can be delivered to the cells in vitro without specific binding and internalization. In our experiments, the effect of activity in the medium was significantly high and also showed an activity-dependent effect on the cell viability. This again stresses the importance of including a non-targeting surrogate to account for medium effects. Both [^153^Sm]Sm-DOTA-TATE as well as [^153^Sm]Sm-DTPA showed decreased viability at high concentrations (>2.5 MBq/mL). At the lowest concentration (1.25 MBq/mL), however, [^153^Sm]Sm-DOTA-TATE significantly lowered viability compared with untreated cells, while this was not the case with [^153^Sm]Sm-DTPA treatment, suggesting that the therapeutic effect of [^153^Sm]Sm-DOTA-TATE is higher than the effect of the medium alone. The effect is, however, limited, as can be attributed to the low A_m_ of the [^153^Sm]Sm-DOTA-TATE.

It is well known that the A_m_ plays an important role in the efficacy of TRNT. Performing experiments with lower-A_m_ radiopharmaceuticals will have consequences for biological evaluation, which is illustrated by the MTS and CFU assays. The low A_m_ in our experiments is, however, caused by the current production method of ^153^Sm. Although the A_m_ of the produced ^153^Sm is rather high, it is currently only produced in relatively low amounts, making it hard to generate sufficient amounts of the radiopharmaceutical. Therefore, the [^153^Sm]Sm-DOTA-TATE in our studies is only obtained in fairly low A_m_ (2–10 MBq/nmol) compared to, respectively, clinical (± 40 MBq/nmol) [17] and preclinical in vitro A_m_ (±50 MBq/nmol) [19] of [^177^Lu]Lu-DOTA-TATE. Efforts are currently underway to increase production efficiency and to ramp up production capacity. This will allow larger scale productions of ^153^Sm, enabling higher-A_m_ radiolabelings. Since we already see distinct cell binding and therapeutic effect with [^153^Sm]Sm-DOTA-TATE at our current low A_m_, this effect is only expected to be greater when higher A_m_ can be reached.

Despite the relatively low A_m_, our in vivo evaluation showed some promising results. In the biodistribution study, a clear and prolonged tumor uptake was observed with a rapid renal excretion, the main excretion pathway of [^153^Sm]Sm-DOTA-TATE. At later time points, a gradual wash out of activity was observed from the tumor tissue. This was comparable to data obtained in a similar animal model [18]. In vivo SPECT-CT images were successfully acquired with the A_m_ achieved and they confirmed the data seen in the biodistribution study at 4 and 24 h p.i. The tumor-to-kidney SUV ratios obtained in our biodistribution studies were remarkably higher compared to [^177^Lu]Lu-DOTA-TATE in a similar animal model [18]. Other organs which express the SSTR_2_, such as the spleen, show a similar uptake of [^153^Sm]Sm-DOTA-TATE compared to [^177^Lu]Lu-DOTA-TATE, including initial high pancreas uptake that was rapidly washed out [18]. Tumor-to-kidney ratios were compared in a AR42J (endogenous expression of SSTR_2_) xenografted mouse model injected with ^55^Co, ^64^Cu and ^68^Ga-labeled DOTA-TATE. [^55^Co]Co-DOTA-TATE had, respectively, a 4- and 8-fold-higher tumor-to-kidney ratio compared to ^64^Cu and ^68^Ga at 24 h p.i., although the initial absolute uptake did not significantly differ. Further preclinical studies on mice will be of interest to investigate the microscopic distribution over time of [^153^Sm]Sm-DOTA-TATE, the resulting absorbed doses within tissues and the associated radiobiological effects (end points of tumor control and toxicity). This will allow assessment of the potential of TRNT with [^153^Sm]Sm-DOTA-TATE in relation to the clinically used [^177^Lu]Lu-DOTA-TATE.

Interestingly, in healthy mice we observed a noticeable retention of [^153^Sm]Sm-DOTA-TATE by the liver and spleen. The expression of SSTRs is highly homologous between mice and humans, and moderate expression of SSTR_2_ is seen in the red pulp of the spleen and has been reported as a site of off-target specific binding [22,23]. SSTR_2_ is expressed in liver to a lesser extent [24] and uptake can be attributed to non-specific binding (metabolic enzymes) or blood circulation and clearance. Uptake in these organs is reduced in tumor-bearing animals. This can be attributed to a ‘sink’ effect influenced by overexpression of SSTR2 in the tumor [22]. This increase in local, high-affinity receptor density will withdraw [^153^Sm]Sm-DOTA-TATE from circulation. Hence, the tumor-to-organ ratio will increase as normally the expression of SSTR_2_ in non-tumor organs remains unchanged. The lower A_m_ of [^153^Sm]Sm-DOTA-TATE can be beneficial in tumor targeting as the fraction cleared by the first pass effect of the liver can be reduced. Part of the high-capacity low-affinity receptors or metabolic enzymes in the liver can be occupied by unlabeled DOTA-TATE (in excess vs. [^153^Sm]Sm-DOTA-TATE), which leads to a higher probability of [^153^Sm]Sm-DOTA-TATE binding to tumor sites [25]. Similar effects can be expected in the kidneys, partially explaining the much higher kidney-to-tissue SUV ratios observed in our experiments. In the earlier time points, pronounced uptake in pancreas was observed in both healthy and tumor-bearing animals, in line with endogenous expression of the SSTR_2_ in this organ [26].

In conclusion, we successfully evaluated, for the first time, high-A_m_ ^153^Sm for TRNT using DOTA-TATE as a vector molecule in vitro and in vivo. Although we radiolabeled at limited A_m_, a clear uptake in tumor cells and a mouse xenograft model clearly indicates the potential of [^153^Sm]Sm-DOTA-TATE in TRNT. This proof-of-concept study using mass-separated isotopes could open doors towards wider application of mass separation in medical isotope production.

## Figures and Tables

**Figure 1 pharmaceutics-14-02566-f001:**
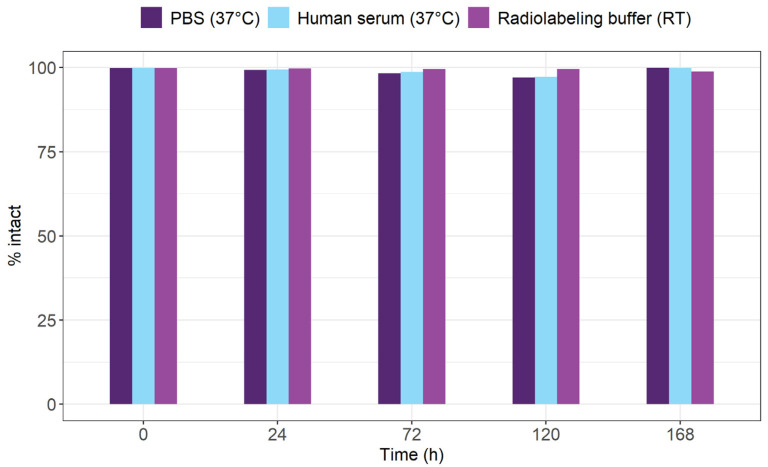
[^153^Sm]Sm-DOTA-TATE did not show any leaching of ^153^Sm from the chelator in PBS at 37 °C, human serum at 37 °C and radiolabeling buffer at RT for at least 7 days. RT = room temperature.

**Figure 2 pharmaceutics-14-02566-f002:**
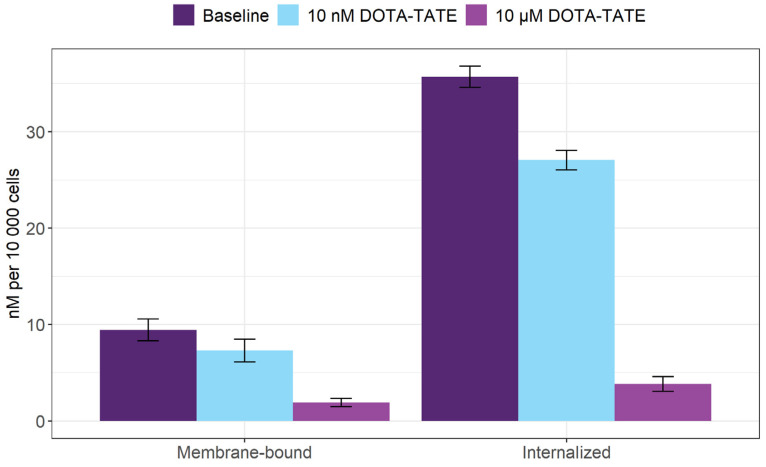
[^153^Sm]Sm-DOTA-TATE is internalized by SSTR_2_-expressing CA20948 cells. Values represent mean ± standard deviation.

**Figure 3 pharmaceutics-14-02566-f003:**
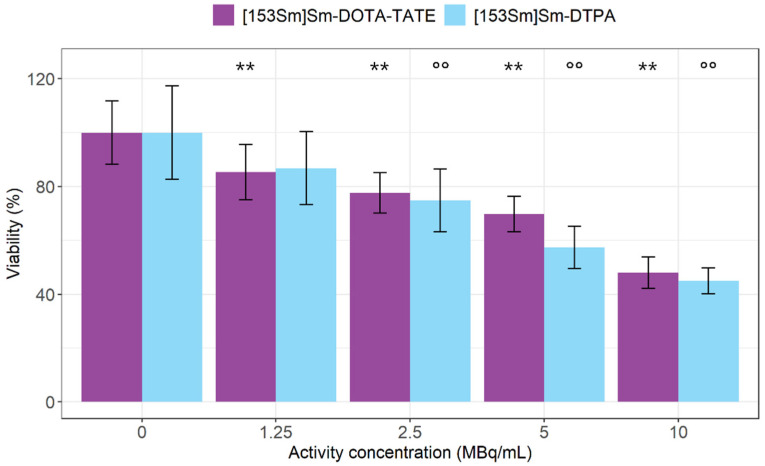
Percentage viability of CA20948 cells after [^153^Sm]Sm-DOTA-TATE treatment (MBq/mL) or [^153^Sm]Sm-DTPA treatment (MBq/mL). Values represent mean ± standard deviation. ** *p* < 0.01 for statistical differences relative to 0 MBq/mL [^153^Sm]Sm-DOTA-TATE-treated cells and °° *p* < 0.01 for statistical differences relative to 0 MBq/mL [^153^Sm]Sm-DTPA-treated cells, both obtained by linear modelling.

**Figure 4 pharmaceutics-14-02566-f004:**
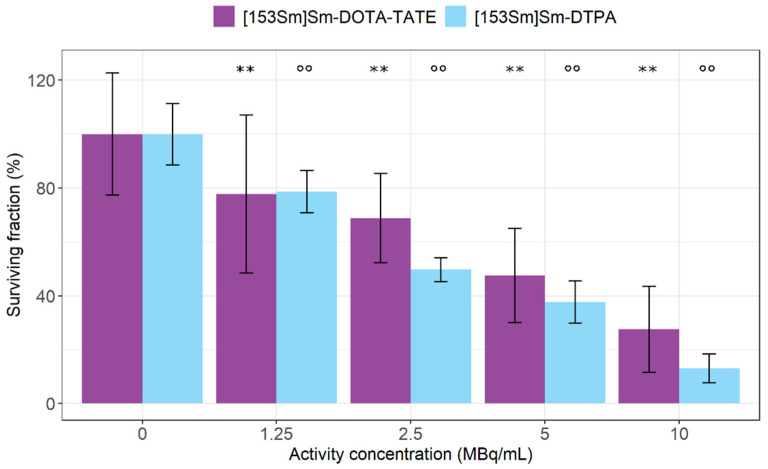
Surviving fraction expressed in percentages of CA20948 cells after [^153^Sm]Sm-DOTA-TATE treatment (MBq/mL) or [^153^Sm]Sm-DTPA treatment (MBq/mL). Values represent mean ± standard deviation. ** *p* < 0.01 for statistical differences relative to 0 MBq/mL [^153^Sm]Sm-DOTA-TATE-treated cells and °° *p* < 0.01 for statistical differences relative to 0 MBq/mL [^153^Sm]Sm-DTPA-treated cells, both obtained by linear modelling.

**Figure 5 pharmaceutics-14-02566-f005:**
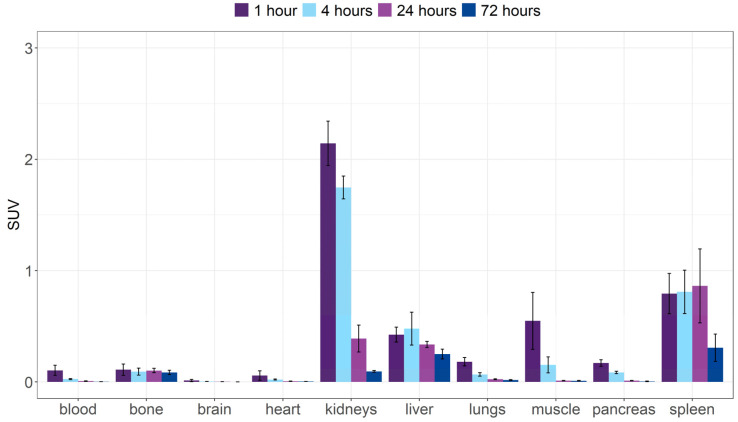
Ex vivo biodistribution over time expressed as standard uptake value of [^153^Sm]Sm-DOTA-TATE in healthy mice. Values represent mean ± standard deviation (*n* = 3–4).

**Figure 6 pharmaceutics-14-02566-f006:**
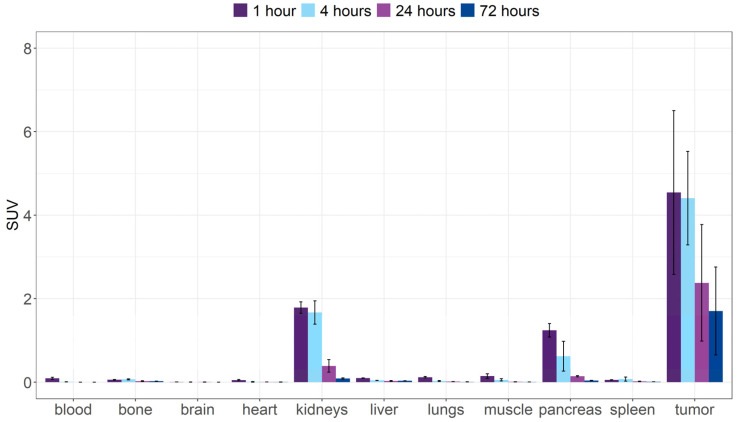
Ex vivo biodistribution over time expressed as standard uptake value of [^153^Sm]Sm-DOTA-TATE in CA20948 xenografted mice. Values represent mean ± standard deviation (*n* = 3–4).

**Figure 7 pharmaceutics-14-02566-f007:**
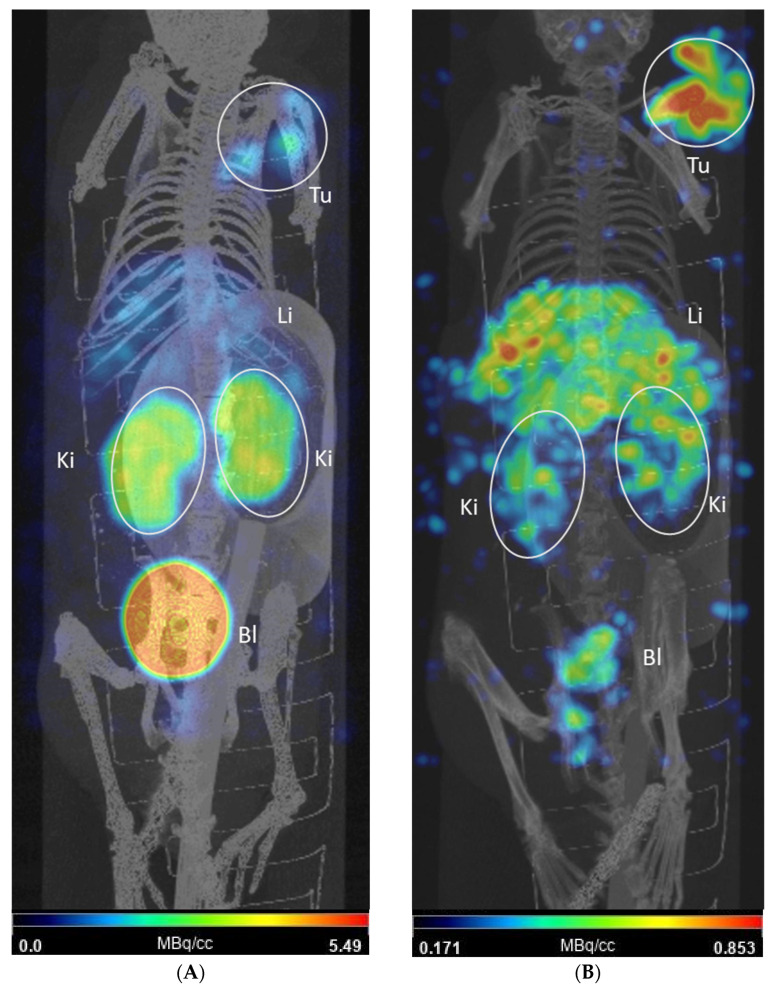
SPECT-CT images shown as maximum intensity projections for a mouse injected with 20 MBq of [^153^Sm]Sm-DOTA-TATE. (**A**): Image obtained at 4 h post injection; (**B**): image obtained at 24 h post injection. The tumor (Tu) can be seen in the right shoulder. Other visible organs are kidneys (Ki), liver (Li) and bladder (Bl).

**Table 2 pharmaceutics-14-02566-t002:** Tumor-to-kidney SUV ratios at different time points post-injection (p.i.) of [^153^Sm]Sm-DOTA-TATE in CA20948 xenografted mice.

Time p.i.	Tumor-to-Kidney
1 h	2.5 ± 1.0
4 h	2.6 ± 0.6
24 h	5.6 ± 2.2
72 h	19.2 ± 10.8

## Data Availability

Not applicable.

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
