# Peer review of "Exploring the Potential of High-Molar-Activity Samarium-153 for Targeted Radionuclide Therapy with [153Sm]Sm-DOTA-TATE"

_pharmaceutics, 2022, doi:10.3390/pharmaceutics14122566_

Round 1

Reviewer 1 Report

The manuscript entitled “Exploring the potential of high-molar activity Samarium-153 for targeted radionuclide therapy with [153Sm]Sm-DOTA-TATE” reports the radiolabelling and first biological assessment of high-molar activity [153Sm]Sm-DOTA-TATE in SSTR2-expressing cells and in tumour models of pancreatic neuroendocrine tumours.

This work builds up on earlier research by the authors (Van de Voorde et al, Front Med 8:675221) that established the potential of combining neutron irradiation with mass separation to provide non-carrier added, high-molar activity Samarium-153 suitable for targeted radionuclide therapy. The current proof-of-concept study demonstrates the potential of mass-separated Samarium-153 for targeted radionuclide therapy and could pave the way towards wider applications of mass separation procedures in the production of medical isotopes.

The manuscript is scientifically sound as well as the methodologies used in the study. The overall level of the composition is very good and in an easily readable style. The experimental detail is sufficient to provide a clear understanding of the procedures. The results are given in sufficient detail as well. The discussion of results is well supported by pertinent literature. The figures and tables are sufficiently clear as well as the SPECT-CT images. The references are thorough and appropriate.

Thus, it is with these considerations in mind that I recommend publication of the manuscript in Pharmaceutics.

However, there are some comments that I would like to address to the authors.

Results. Page 9. The small-animal SPECT-CT scans in tumour-bearing mice resulted in good quality images. However, I would suggest that the authors could identify in the image the organs with more activity for sake of better understanding. Also, the authors could detail the discussion about the SPECT-CT images a little bit.

Discussion. Page 12. The authors claim that liver and spleen uptake is reduced in tumour bearing animals, probably due to a “sink” effect of the tumour. However, this issue is not clearly explained (line 419-422).  Can the authors make these two paragraphs more understandable for the reader?

Author Response

Point by point response attached

Reviewer 2 Report

This work presents the ability of Samarium-153 for targeted radionuclide therapy with [153Sm]Sm-DOTA-TATE in in vivo study. In my opinion, the manuscript is suitable for publication in Pharmaceutics. My comments are as follows:

1.       The authors should give the table for the performance comparison of Samarium-153 with other theranostic radionuclides.

2.       The texts in all figures are quite small and hard to read (except figure 7).

3.       The methods obtained from published papers need to be cited.

Author Response

Point by point response attached

Reviewer 3 Report

The manuscript submitted to the special issue “Radiopharmaceuticals for Cancer Imaging and Therapy” to pharmaceutics by Vermeulen and co-workers examines the potential of using 153Sm of high molar activity for radiopharmaceutical purposes in theranostic contexts. It is the continuation of a previous work describing mainly the production and work-up procedures of high Am 153Sm. This time, 153Sm was radiolabeled with DOTATATE which is well-known in clinics and already approved in combination with Lutetium-177. The authors as well as some literature studies reveal that 153Sm might have some advantages over 177Lu by emitting higher energy beta particles and Auger electrons. Moreover, 153Sm emits photons around 100 keV, thereby making it a suitable SPECT nuclide using standard clinical setup. Although 153Sm will not be considered for solely diagnostic use it is a benefit to follow therapeutic effects in patients over a period of several days, which might work better for 153Sm compared to 177Lu du to the lower photon energy. All in all, I rate the work as highly interesting for the radiopharmaceutical and nuclear medicine community. The use of high Am ­153Sm is novel and the comparison to 177Lu-conjugates will be necessary to get 153Sm more into the clinics. I congratulate the authors to their interesting findings regarding the cell damaging potential of 153Sm-DOTATATE as well as for the promising biodistribution data. The manuscript’s style and experimental data are written in a very clear manner, nonetheless, some aspects are missing. I conclude that the manuscript is suitable for publication but encourage the authors to address the following aspects before submitting a revised version of the manuscripts.

Major comments:

·         Regarding Figure 1 and the surrounding paragraphs. Have you somehow checked for metabolic products when talking about stability of your radiolabeled complexes in biologically relevant media? There are many methods known such as precipitation methods to evaluate possible metabolites afterwards (by HPLC for example). In general, HPLC would be maybe more suitable to follow your conjugate’s integrity over time. Just to show, that the 153Sm stays in the DOTA (which is the only thing you can state by checking a single TLC stripe in citrate) is too weak to assume that everything is totally stable – at least you should minimize your statement to the binding capacity of DOTA for 153Sm.

·         Regarding your Figures 3 and 4: Even if you have discussed this issue, for me it is not clarified enough, why 153Sm-DTPA is so harmful to your cells. In my mind, there are some options/experiments you can consider for the validation of your data and the following conclusions. I suggest either

o   a) to check for the binding of the DTPA-complex to this certain cell line you have used, or

o   b) to check by additionally incubating the cells with a radical scavenger to see if there are differences (which should be the case if DOTATATE is binding and DTPA just “flying around”) or

o   c) to check for “free” 153Sm as an additional reference compound to see if there are differences compared to 153Sm-DTPA.

·         How does the 153Sm-TATE look like in bd studies compared to other radiometal complexes? You can comment on this in the discussion section.

·         Why is the pancreas SUV low in the healthy mice (Fig5), but seems to be significantly higher in your xenografted studies? Endogenous expression should be similar.

·         Assuming, that the majority of your radiotracer is internalized in the tumor cells: Why do you observe this intense activity decrease in your tumor model? Do you have any idea?

Minor comments:

·         Line 78: Full stop is missing

·         Line 317: “mice” should be “mouse”

Author Response

Point by point response attached

Reviewer 4 Report

The article is based on a very comprehensive work involving neutron activation and mass separation of Sm-152, radiolabeling and the experimental work and analysis of the effect of treatment of cancer in mice. The reviewer background is more toward nuclear science, however impressed with the care of the authors to document every aspects of their work. 

The main point of the paper to highlight the potential of the use of the beta emitter Sm-153 for targeted cancer treatment. One of the important result of their present and previous work (ref [6]) was to develop a method to produce high specific activity of Sm-153 by combining neutron irradiation and mass separation. This isotope has advantage over the commonly used Lu-177, namely the larger number of low energy Auger and conversion electrons. On page 1 ref [1] was used as the source of nuclear decay data. However ref [1] only contains mean energies and no information on intensities. More detailed information is available at the LiveChart at the IAEA web site. The mean energy of the Auger electrons from the decay of Sm-153 is around 5.5 keV and on average, 6.6 Auger electrons are emitted per decay. 

On page 5 the authors report, that S values are calculated using the MC code MCNP6.2, without referring to the source of the radiation spectrum. It would be very useful to report these details.

Author Response

Point by point response attached

Round 2

Reviewer 2 Report

The quality and presentation of this manuscript have been enhanced after the revision round 1. The authors have indicated the reviewer's comments properly. I recommend this manuscript for the publication on Pharmaceutics journal. 

Reviewer 3 Report

All comments have been addressed - suitable for publication in current form.